# Laryngopharyngeal Reflux Scoring in a Pediatric Population

**DOI:** 10.3390/jcm12237425

**Published:** 2023-11-30

**Authors:** Ivan Abičić, Marina Čović, Milorad Zjalić, Marina Bakula, Ksenija Marjanović, Anamarija Šestak, Branko Dmitrović, Tihana Mendeš, Martina Smolić, George Y. Wu, Hrvoje Mihalj, Željko Zubčić, Andrijana Včeva

**Affiliations:** 1Department of Otorhinolaryngology and Maxillofacial Surgery, Faculty of Medicine, Josip Juraj Strossmayer University of Osijek, 31000 Osijek, Croatia; tihanamendes811@gmail.com (T.M.); hrvoje.mihalj@gmail.com (H.M.); zzubcic21@gmail.com (Ž.Z.); andrijana.vceva@gmail.com (A.V.); 2Department of Otorhinolaryngology and Head and Neck Surgery, Clinical Hospital Centre Osijek, 31000 Osijek, Croatia; 3Department of Pharmacology and Biochemistry, Faculty of Dental Medicine and Health, Josip Juraj Strossmayer University of Osijek, 31000 Osijek, Croatia; marinacovic19@gmail.com (M.Č.); martina.smolic@fdmz.hr (M.S.); 4Department of Molecular Medicine and Biotechnology, Faculty of Medicine, University of Rijeka, 51000 Rijeka, Croatia; 5Department for Pathology and Forensic Medicine, Clinical Hospital Centre Osijek, 31000 Osijek, Croatia; bakula.marina@gmail.com (M.B.); marjanovic.ksenija@kbco.hr (K.M.); dmitrovic.branko@kbco.hr (B.D.); 6Department of Anatomy, Histology, Embryology, Pathological Anatomy and Pathological Histology, Faculty of Dental Medicine and Health, Josip Juraj Strossmayer University of Osijek, 31000 Osijek, Croatia; 7Department of Medicine, Division of Gastroenterology/Hepatology, University of Connecticut Health Center, Farmington, CT 06030, USA; wu@uchc.edu

**Keywords:** reflux symptom index, reflux finding score, pepsin, saliva, palatine tonsil, laryngopharyngeal reflux, tonsillectomy

## Abstract

In recent years, the prevalence of laryngopharyngeal reflux has risen, especially among pediatric patients. The diagnosis of laryngopharyngeal reflux relies on patient history and clinical assessment using the Reflux Finding Score and Reflux Symptom Index as crucial diagnostic tools. Some studies have proposed a link between pepsin and laryngopharyngeal reflux, potentially triggering palatine tonsil hypertrophy. Our study aimed to investigate the correlation between laryngeal and pharyngeal manifestations of laryngopharyngeal reflux through two questionnaires and the presence of pepsin in saliva and palatine tonsils in a pediatric population. Pepsin in saliva was detected using a Western blot method, while immunohistochemistry assessed its presence in palatine tonsils. Although no statistically significant differences in Reflux Finding Score and Reflux Symptom Index were found between the immunohistochemistry-positive (IHC-positive) and immunohistochemistry-negative (IHC-negative) groups, median reflux symptom index and Reflux Finding Score values consistently trended higher in the IHC-positive group. This suggests a potential connection between elevated index values and pepsin presence in tonsillar tissue. Further investigations are essential to fully comprehend the clinical implications of these findings.

## 1. Introduction

Laryngopharyngeal reflux or extraesophageal reflux is a condition characterized by the regurgitation of gastric contents into the mucosal area of the larynx and pharynx [1]. Over the past fifty years, the reported prevalence of laryngopharyngeal reflux has increased by 4% annually [2] in both adults and the pediatric population [3,4]. Symptomatic laryngopharyngeal reflux presents with an irritating cough, laryngospasm, dysphagia, postnasal drip, and laryngitis [5]. It was earlier considered that laryngopharyngeal reflux and gastroesophageal reflux disease are synonyms, but there are significant differences between these two conditions. While gastroesophageal reflux disease usually appears as a consequence of lower esophageal sphincter weakness, laryngopharyngeal reflux is characterized by both upper and lower esophageal sphincter weakness [6]. Pepsin is an endopeptidase produced by the chief cells in the stomach in its inactive form, pepsinogen, with a mass of around 42 kDa. Upon contact with stomach acid, part of the protein is autocatalytically cleaved. The cleaving process produces pepsin, which is an active form of the enzyme with a mass of around 36 kDa [7,8,9]. Pepsin is most active at pH values between 1.5 and 2 and becomes inactive at pH values greater than 6.5. However, pepsin remains in its stable form even at pH values around 8, and can be reactivated in a more acidic environment [10,11]. Chronic exposure of pepsin to the laryngeal mucosa can contribute to dysplasia and laryngeal carcinoma. In conditions like laryngopharyngeal reflux, pepsin can be found in the upper and lower respiratory tracts, where it can disrupt local defense mechanisms, increase oxidative stress, and activate inflammatory cytokines [12,13,14]. Local defense mechanisms include upper and lower esophageal sphincters, the motility of esophageal musculature, and resistant esophageal mucosa. Specifically, two enzymes called carbonic anhydrase isoenzyme III and squamous epithelium stress protein 70 have complementary roles to local defense mechanisms. Carbonic anhydrase isoenzyme III influences local alkalinization and consequent deactivation of certain components of the stomach contents, while squamous epithelium stress protein 70 has a regulatory effect on the cellular passage of proteins. The main factor in weakening this physiological barrier is pepsin. Through its action, pepsin disrupts intercellular connections and directly reduces the quality of the ciliary function of the mucous membrane of the upper respiratory tract. Namely, it is precisely carbonic anhydrase isoenzyme III and squamous epithelium protein 70 that are the main targets of pepsin’s action, and their levels are drastically reduced in the presence of pepsin [15,16,17]. The diagnosis of laryngopharyngeal reflux heavily relies on patient history and clinical examination of the laryngeal and pharyngeal mucosa. However, for a more comprehensive assessment of laryngopharyngeal reflux symptoms, Belafsky and colleagues developed the Reflux Symptom Index questionnaire [16] and the Reflux Finding Score scale [17]. Although 24 h pH monitoring is considered the gold standard for diagnosing laryngopharyngeal reflux, the Reflux Finding Score and Reflux Symptom Index have proven to be practical and reliable diagnostic tools as well as tools for monitoring the response to laryngopharyngeal reflux therapy [5,18]. Palatine tonsils are lymphoepithelial organs located in the oropharynx and play a significant role in defending the human body against various microorganisms, reaching their peak immunological activity in the first decade of human life [19,20]. They are located posterior to the hard palate just below and behind the uvula, positioned between the anterior and posterior faucial pillars. The anterior faucial pillar, also called the palatoglossal arch, extends from the uvula to the base of the tongue, defining the boundary between the oral cavity and the oropharynx. On the other hand, the posterior faucial pillar, known as the palatopharyngeal arch, stretches from the uvula downward to the side of the throat, further delineating the separation between the oral cavity and oropharynx (Figure 1A). Due to their characteristic location at the intersection of the gastrointestinal system and upper respiratory tract, they are considered to be crucial immunocompetent tissue in the human body [21]. Palatine tonsils can histologically be divided into four compartments: lymphoepithelium, germinal centers of lymphatic follicles, mantle zone, and the area between lymphatic follicles [22] (figure in Section 3). They are covered by a stratified squamous epithelium, which deeply invaginates into the lymphatic tissue of the palatine tonsils, forming between 15 and 30 crypts. The epithelium of tonsillar crypts initially serves as the point of contact between the body and antigens. Beneath the stratified squamous epithelium of the palatine tonsils are fenestrated capillaries and lymphatic tissue with numerous lymphatic follicles, where lymphocytes come into contact with the epithelial cells. Macrophages, dendritic cells, and high endothelial venules within the lymphatic tissue also play a significant role in initiating an immune response to external antigens. In a certain number of individuals, especially in children aged between 3 and 6 years, there is hypertrophy of the palatine tonsils. Depending on the severity of the hypertrophy of the Waldeyer’s lymphatic ring tissue, this condition can lead to middle ear inflammation, hearing loss, snoring, sinusitis, cough, frequent nighttime awakenings, frequent colds, and swallowing difficulties in pediatric and adult populations [23]. Although earlier research focused on the bacterial etiology of palatine tonsil tissue hypertrophy, recent studies have shifted their primary focus to the action of pepsin and extraesophageal reflux as potential triggers for these changes in palatine tonsil tissue [13,24]. A study by Kim and colleagues in 2016 proposed two hypotheses regarding the possible mechanism of tonsillar hypertrophy due to pepsin action. The first hypothesis involves direct contact between pepsin and tonsillar lymphocytes, which subsequently proliferate and influence the growth of lymphatic follicles and tonsillar hypertrophy. The second hypothesis involves the action of macrophages, whose activation in contact with pepsin leads to the secretion of cytokines and an inflammatory reaction that results in tonsillar hypertrophy [25]. The same group of researchers in 2018 investigated the in vitro effects of pepsin on tonsillar hypertrophy and concluded that the number of CD4-positive cells from hypertrophic tonsil tissue significantly increased in the presence of pepsin and that the pediatric population is more sensitive to pepsin exposure compared to adults [26].

## 2. Materials and Methods

### 2.1. Ethics Statement

The study was conducted with the approval of the Ethics Committees at both the Osijek Clinical Hospital Center and the Faculty of Medicine Osijek, University of Josip Juraj Strossmayer Osijek (R1/13151/2021). We strictly adhered to the ethical principles set forth in the Nuremberg Code and the Declaration of Helsinki. All patients included in the study were under the age of 18. 

### 2.2. Study Subjects and Questionnaires

The study started on 16 June 2021 and ended on 15 September 2023. The study included 76 pediatric patients at the Osijek Clinical Hospital Center with an indication for tonsillectomy with certain inclusion and exclusion criteria (Table 1). Inclusion criteria were age under 18, thorough medical history and palatine tonsils hypertrophy confirmed by physical examination, and Brodsky tonsil grading score of 2+ and more. Brodsky grading scale is classified into five grades: grade 0 implies previous tonsillectomy, grade 1 implies that tonsils are hidden within pillars, grade 2 implies that tonsils are beyond anterior pillar and that they occupy 25–50% of pharyngeal space, grade 3 indicates that tonsils are beyond the pillars and occupy 50–75% of pharyngeal space, and grade 4 indicates that tonsils occupy more than 75% of pharyngeal space [27]. Exclusion criteria were positive anti-streptolysin titer and contraindications for tonsillectomy, such as systemic disorders and other clinical conditions.

Indications for tonsillectomy include the presence of obstructive sleep apnea in children and frequent tonsillitis of more than 7 per year or 5 tonsillitis per year for two years. Tonsillectomies were performed under general anesthesia, with tissue sampling from the inferior tonsil area. No postoperative complications were reported. Samples of palatine tonsils were fixed in neutral buffered 4% formaldehyde solution (Biognost, Zagreb, Croatia), dehydrated in ascending alcohol series (Biognost, Zagreb, Croatia), clear in xylene (Sigma-Aldrich, Saint Louis, Missouri, United States) and embedded in histological paraffin wax (Merck, Darmstadt, Germany). Saliva samples taken before surgery were stored in transport tubes and stored in a refrigerator at −80 °C after the addition of 10× solution of complete mini protease inhibitors (Roche, Basel, Switzerland). Tonsil tissue was used for immunohistochemical detection of anti-Pepsin A, while saliva samples were used for Western blot analysis of anti-Pepsin A.

Based on a detailed otorhinolaryngological examination, the Reflux Finding Score questionnaire was filled out for each patient. The Reflux Finding Score scale comprises 8 clinical findings in the larynx, documented by endovideolaryngoscopic examination by a physician: subglottic edema, ventricular obliteration, mucosal hyperemia, vocal cord edema, diffuse laryngeal edema, posterior commissure hypertrophy, laryngeal granuloma, and dense endolaryngeal mucus. A total score exceeding 7 out of a possible 26 points reliably confirms the presence of laryngopharyngeal reflux [15,16,17]. Reflux Symptom Index questionnaire assesses the intensity of nine different symptoms: hoarseness, throat clearing, sensation of increased throat mucus or postnasal drip, difficulty swallowing, cough after eating in a seated position, dyspnea, sensation of a lump in the throat, and heartburn. A total score exceeding 12 out of a possible 45 points raises strong suspicion of laryngopharyngeal reflux. Each patient or legal guardian filled out the Reflux Symptom Index questionnaire. Both questionnaires were completed on the day of surgical procedure. The data were entered into the database created for this research. Data entry was performed during the research, and the database was regularly stored with the creation of backup copies.

### 2.3. Western Blot Analysis of Pepsin A in Salivas

Before electrophoresis, total proteins in the saliva samples were determined using the Bradford method in microplate flat-bottom wells following the manufacturer’s instructions (Biorad, Hercules, CA, USA) and results were used for sample normalization. In addition to the subjects’ samples, prestained protein ladder—mid-range molecular weight (Abcam, Cambridge, UK) and standard samples of 1 µg and 0.1 µg of porcine gastric mucosa pepsin A (sc-271798, Santa Cruz Biotechnology, Portland, OR, USA) were used in the first, second, and third well, respectively. Hoeffer SE250 Western blot system with casted 1.5 mm 12% Bis-Tris gels with 1.25% 2,2,2-trichloroethanol for stain-free detection was used to run samples at 25 mA per gel. Samples were blotted to PVDF membrane (Merck) in Towbin buffer with 20% methanol using TE22 Mini Tank Blotting Unit (Hoeffer inc. San Francisco, CA, USA). Subsequently, the membranes were incubated in a blocking solution comprising 3% *m*/*v* of bovine serum albumin (BSA) (Sigma-Aldrich, Saint Louis, MI, USA) and in PBS buffer with 0.1% Tween (PBS-T) for one hour at room temperature. In the next step, the membranes were incubated in the primary antibody solution for up to 48 h with rotation at 4 °C. The primary antibody used was anti-Pepsin A at a 1:500 dilution (sc-271798, Santa Cruz Biotechnology, Dallas, TX, USA). After incubation in the primary antibody solution, the membranes were washed in 1×PBS-T buffer, followed by incubation in the appropriate secondary antibody solution for two hours at room temperature with rotation. Subsequently, the membranes were rinsed again in 1×PBS-T, then incubated in the corresponding developer solution (Immun-Star WesternC Kit, Bio-Rad, Hercules, CA, USA), and visualized using a visualization system ChemiDocTM Imaging system (Bio-Rad, Hercules, CA, USA). The obtained results were analyzed and quantified using the ImageJ/FIJI software (National Institutes of Health, Maryland, United States).

### 2.4. Immunohistochemical Detection of Pepsin in Tonsillar Tissue

The procedure was performed at the Clinical Department of Pathology and Forensic Medicine at the Clinical Hospital Center Osijek using the VENTANA Bench Mark Ultra system, following the manufacturer’s protocol. When inputting the anti-Pepsin A antibody (sc-271798, Santa Cruz Biotechnology, Dallas, TX, USA) procedure into the Ventana Bench Mark Ultra system, its protocol was entered into the device. Each sample was barcode labeled. The paraffin-embedded tissue 2 section on the glass slide was inserted into the device, and it underwent a fully automated deparaffinization process using EZ Prep concentrate (LOT 206236-01, REF 950-102), which was diluted at a ratio of 1:10, placed in a container, and then applied to the slides using the device. The detection of Pepsin A was performed using the Ultra View Universal DAB Detection kit (REF 760-500, LOT G07369) following the manufacturer’s protocol. After deparaffinization, the slides were rinsed with Reaction Buffer, which was diluted by mixing 2 L of the reaction buffer with 18 L of distilled water (REF 950-300, LOT D05414). The diluted buffer was placed in a container from which the device withdrew the solution. Endogenous peroxidase inhibition was carried out using Ultra View Universal DAB Inhibitor (3% H_2_O_2_) for 10 min, followed by rinsing in Reaction Buffer. Antigen retrieval in tissue, as a potential binding site, was achieved by incubating with Ultra CC1 (Ultra Cell Conditioner 1, pH 8.4) for 30 min, followed by rinsing in Reaction Buffer. The anti-Pepsin A antibody used was a mouse monoclonal antibody from Santa Cruz Biotechnology, diluted at a ratio of 1:100. Incubation lasted for 40 min at room temperature. The labeled sites of the immunoreaction were visualized using Ultra View DAB Chromogen and Ultra View Universal DAB Copper for 5 min. After staining, the slides were rinsed with Reaction Buffer. Hematoxylin staining was performed using an automated system from the Ventana kit. The slides were rinsed with distilled water, rehydrated through a series of alcohol baths from lower to higher concentrations, and manually cleared in a xylene substitute outside the device [28]. After clearing from the xylene substitute, the slides were covered with an automatic cover in the Dako Sakura system, covered with Sakura Tissue-Tek Cover Slipping Film (LOT 4840210). The slides were examined under an OLYMPUS BX46 microscope (Olympus, Tokyo, Japan) at a magnification of 200×. A positive control was a human stomach in which the aforementioned positivity was much more pronounced and intensively expressed in all cellular and muscular structures.

### 2.5. Statistical Analysis

Sorted data were tested for normality of distribution with the Shapiro–Wilk test [29,30]. Data were not normally distributed, and the non-parametric Mann–Whitney U test was performed to assess potential significant differences between groups. For the nonrandom association of Reflux Finding Score or Reflux Symptom Index with immunohistochemistry (IHC), Fisher’s exact test was used. For all statistical tests, the significance level (α) was set at 0.05. Statistical analysis was performed using Statistica 13 statistical software (TIBCO, Palo Alto, CA, USA).

## 3. Results

Immunohistochemical analysis was used to assess pepsin presence in palatine tonsil tissue. Of the samples, 35 were negative, showing no reaction to pepsin, while 41 were positive for pepsin. Human pyloric tissue served as a positive control for validation (Figure 2A). In the positive samples, a majority of staining was localized in the cells of the germinal center. Pepsin staining was predominantly observed in the connective tissue adjacent to the lymphoid tissue and in the marginal zones. These results were the foundation for the categorization of data into two groups: immunohistochemical-positive (IHC-positive) (Figure 2B) and immunohistochemical-negative (IHC-negative) (Figure 2C).

Analyzing IHC-positive and IHC-negative groups did not yield statistically significant differences between the Reflux Finding Score and Reflux Symptom Index. However, median values for the IHC-positive group were consistently higher, suggesting a potential association between higher index values and the presence of pepsin on tonsillar tissue (Figure 3A,B).

Western blot analysis of saliva samples did not reveal significant differences in pepsin levels between the IHC-positive and IHC-negative groups. Pepsin signals were weak and primarily detected at around 36 kDa. The use of a pepsin protein standard as the first two samples helped mitigate issues regarding the pepsin band position in salivary samples (Figure 4).

Two-Way ANOVA analysis of gender and age effects on positive or negative findings of pepsin in tonsillar tissue showed no meaningful associations between those three parameters (Table 2). Also, there is no significant variability in age between genders, which makes relatively homogenous groups age-wise.

The Fisher exact test showed no meaningful associations between Reflux Finding Score scores and tonsil tissue staining. However, Reflux Symptom Index scores were significantly associated with pepsin staining. A Reflux Symptom Index score of 13 and above was indicative of the probable presence of pepsin on tonsillar tissue (Table 3 and Table 4).

## 4. Discussion

Even though there is no precise information about laryngopharyngeal reflux prevalence in the pediatric population, it is presumed to be most present in the first year of life, mainly due to immature esophageal sphincters [31]. Previously, laryngopharyngeal reflux and gastroesophageal reflux disease were thought to be synonymous, but there are significant differences between the two conditions. Namely, the occurrence of laryngopharyngeal reflux is a consequence of the weakness of the upper and lower esophageal sphincter, while in gastroesophageal reflux disease, there is a weakness of the lower esophageal sphincter. Furthermore, although it plays a negligible role in the pathophysiology of gastroesophageal reflux disease, pepsin is an essential factor for laryngopharyngeal reflux, unlike other components of the stomach contents such as bile and acids, which do not always have to be present. What is more, laryngopharyngeal reflux clinical symptoms, presentation, and therapeutic response seem to be very different from gastroesophageal reflux disease, especially in the pediatric population [6,32]. The prevalence of gastroesophageal reflux disease in the pediatric population seems to be growing rapidly, with up to 75% of children with symptoms such as dysphonia having been diagnosed with gastroesophageal reflux disease [33]. On the other hand, there is scarce evidence of laryngopharyngeal reflux prevalence in the pediatric population, mostly due to the diversity of the symptoms and limited diagnostic tools [34]. Moreover, Leichen states in his review that older children tend to have more distinct laryngopharyngeal reflux symptomatology rather than gastroesophageal reflux disease symptomatology [34]. The pediatric population has shown increased sensitivity to the presence of pepsin compared to adults [26]. Laryngopharyngeal reflux symptomatology impacts the patient’s quality of life and presents a diagnostic and therapeutic challenge to physicians [35]. In addition to patient history and clinical examination, the diagnosis of laryngopharyngeal reflux relies on the Reflux Finding Score and Reflux Symptom Index. The advantages of the Reflux Finding Score and Reflux Symptom Index are in their simplicity and accessibility, although these clinical questionnaires have limitations [36,37]. Many recent studies have highlighted pepsin as a potential diagnostic marker for laryngopharyngeal reflux, with several options on how to detect pepsin. Features like accessibility, low price, and noninvasiveness prove that salivary pepsin tests could be a suitable option for laryngopharyngeal reflux diagnosis. On the other hand, some studies suggest that biopsy tests of laryngeal and hypopharyngeal mucosa have higher sensitivity than salivary tests, but also tend to be more aggressive methods in comparison to saliva or sputum collection [38,39]. A Study by Divakaran et al. suggests that combination of positive salivary pepsin test, Reflux Finding Score, Reflux Symptom Index questionnaires, and successful PPIs treatment response strongly increases chances of laryngopharyngeal reflux presence [40,41,42]. On the other hand, Bobin and colleagues emphasize that the concentration of pepsin in saliva does not reflect the actual concentration of active pepsin in the throat and laryngeal tissue [43]. In our study, saliva samples were collected from participants in the morning upon arrival for the surgical procedure. Participants abstained from food and beverages and refrained from oral hygiene for 10–12 h before saliva sample collection. While the study by Na and colleagues suggests that pepsin concentrations in saliva are highest upon awakening, numerous other studies have shown varying results depending on the timing and frequency of saliva sample collection [41,44]. What is more, taking multiple saliva samples throughout the day could increase the salivary pepsin test sensitivity [45]. Although there were no statistically significant differences in Reflux Finding Score and Reflux Symptom Index between IHC-positive and IHC-negative groups, our study suggests a potential association between elevated Reflux Finding Score and Reflux Symptom Index values and the presence of pepsin in the palatine tonsil tissue. Wang and colleagues demonstrated significantly higher Reflux Finding Score and Reflux Symptom Index values in the pepsin-positive group compared to the pepsin-negative group. Spyridoulias and colleagues’ research indicated that pepsin concentration in saliva was associated with laryngeal symptoms in patients assessed using the Reflux Finding Score [46]. Jung and colleagues did not find significant differences in Reflux Finding Score and Reflux Symptom Index between the pepsin-positive and pepsin-negative groups. However, the results of their study suggest a potential simultaneous influence of multiple components of gastric content in the development of laryngopharyngeal reflux. Furthermore, Bobin and colleagues believe that the impact of other components of gastric content in laryngopharyngeal reflux may be the reason why pepsin-negative individuals exhibit clear laryngopharyngeal reflux symptoms [43,47]. Lastly, Sereg-Bahar and colleagues’ study showed a significant correlation between Reflux Finding Score and Reflux Symptom Index values and pepsin and bile acid levels in participants’ saliva [48].

## 5. Conclusions

In conclusion, these findings highlight the relationship between the presence of pepsin in tonsillar tissue and clinical parameters, providing insights into its potential significance in reflux-related conditions. Further investigations are necessary to fully understand the clinical implications.

## Figures and Tables

**Figure 1 jcm-12-07425-f001:**
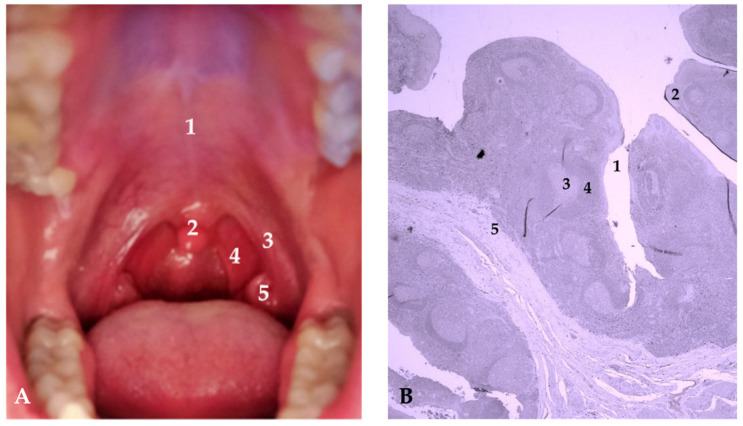
(**A**) Anatomical position of the palatine tonsils: 1—hard palate; 2—uvula; 3—anterior faucial pillar; 4—posterior faucial pillar; 5—palatine tonsil. (**B**) Histological section through the palatine tonsil: 1—crypt; 2—lymphoepithelium; 3—germinal center of lymphatic follicles; 4—mantle zone; 5—area between lymphatic follicles. Magnification 200×; scale 50 µm.

**Figure 2 jcm-12-07425-f002:**
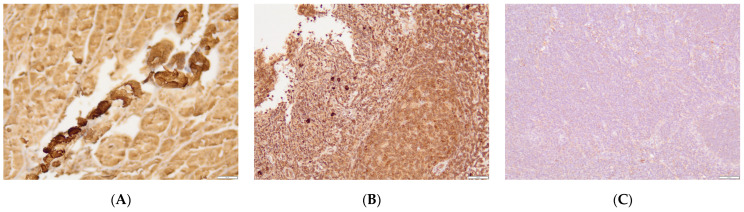
Immunohistochemistry of tonsil tissue stained against pepsin and visualized with DAB-HRP reaction. Positive staining was mostly visualized around germinal centers, connective tissue, and marginal zones. (**A**) Human stomach as positive control; (**B**) tonsil tissue—positive staining; (**C**) tonsil tissue—negative staining. Magnification 200×; scale 50 µm. A positive control was a human stomach in which positive reaction was much more pronounced in all cellular and muscular structures.

**Figure 3 jcm-12-07425-f003:**
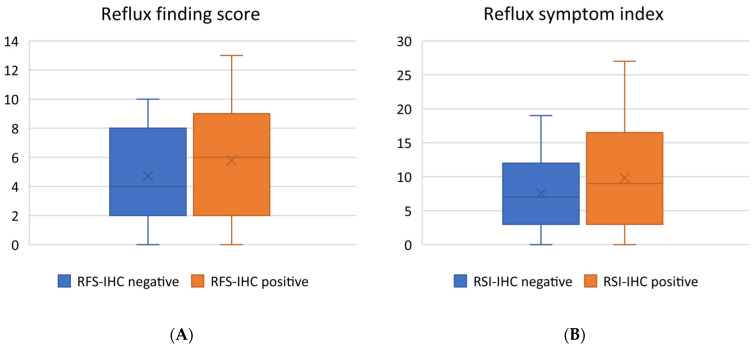
Box and whisker plots representing (**A**) the Reflux Finding Score (RFS) and (**B**) the Reflux Symptom Index (RSI). Statistical analysis was performed using the Mann–Whitney U test to assess potential significant differences, with a significance level (α) set at 0.05. The results of the Mann–Whitney U test revealed the following: for RFS, the U statistic (URFS) = 594.5, *p*-value (*p*) = 0.20174; for RSI, the U statistic (URSI) = 598.5, *p*-value (*p*) = 0.21686.

**Figure 4 jcm-12-07425-f004:**
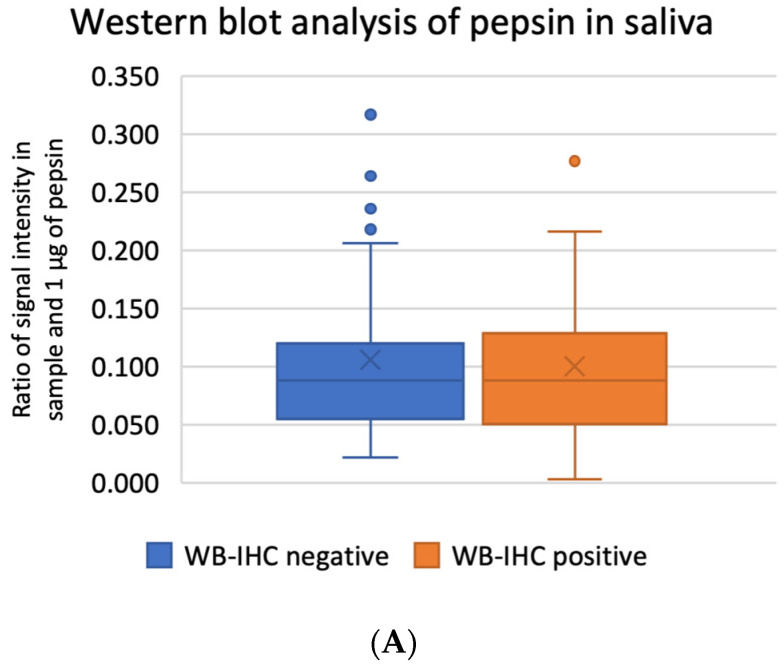
Quantitation of immunoblotting assays of pepsin in saliva. (**A**) Box and whisker plots representing the percentage of pepsin signal calculated against 1 µg of pepsin standard loaded in the first well. Statistical analysis was performed using the Mann–Whitney U test to assess potential significant differences, with a significance level (α) set at 0.05. The results of the Mann–Whitney U statistic (Uwb) = 716.5, *p*-value (*p*) = 0.99584. (**B**) Representative Western blot membrane of immunoblotting against pepsin in saliva. The first two samples are porcine pepsin standards, followed by saliva samples. Signal intensity normalization was performed using stain-free blots.

**Table 1 jcm-12-07425-t001:** Inclusion and Exclusion criteria.

Inclusion Criteria	Exclusion Criteria
age < 18 years of age palatine tonsils hypertrophy confirmed by physical examination and according to medical history Brodsky tonsil grading scale > 2	positive anti-streptolysin titer contraindications for tonsillectomy, such as systemic disorders and other clinical conditions

**Table 2 jcm-12-07425-t002:** Age and gender distribution in regard to immunohistochemistry (IHC) findings.

Gender	Count		Age Average (Years)	Age −95% CI ^1^	Age +95% CI	Two-Way ANOVA Results
Male	IHC negative	18	6.88	5.54	8.23	F(_1,72_) = 0.08602; *p* = 0.7701
	IHC positive	23	7.87	6.45	9.29	
Female	IHC negative	17	7.00	5.42	8.57	
	IHC positive	18	7.55	5.86	9.25	

^1^ CI = Confidence interval.

**Table 3 jcm-12-07425-t003:** Nonrandom association of Reflux Finding Score (RFS) and immunohistochemistry (IHC) finding.

IHC Finding	Count or % Relative to RFS Result			
	RFS Below 7	RFS 7 and Above	Total	*p* ^1^
IHC negative	21 (27.63%)	14 (18.42%)	35 (46.05%)	0.49
IHC positive	21 (27.63%)	20 (26.32%)	41 (53.95%)	
Total	42 (55.26%)	34 (44.74%)	76 (100%)	

^1^ Fisher’s exact test.

**Table 4 jcm-12-07425-t004:** Nonrandom association of reflux symptoms index (RSI) and immunohistochemistry (IHC) finding.

IHC Finding	Count or % Relative to RSI Result			
	RSI Below 13	RSI 13 and Above	Total	*p* ^1^
IHC negative	28 (36.84%)	7 (9.21%)	35 (46.05%)	0.02
IHC positive	22 (28.95%)	19 (25%)	41 (53.95%)	
Total	50 (65.79%)	26 (34.21%)	76 (100%)	

^1^ Fisher’s exact test.

## Data Availability

Data are available upon request due to privacy restrictions. The data presented in this study are available on request from the corresponding author. The data are not publicly available due to minor patients’ privacy.

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
