# Peer review of "Laryngopharyngeal Reflux Scoring in a Pediatric Population"

_jcm, 2023, doi:10.3390/jcm12237425_

Round 1

Reviewer 1 Report

Comments and Suggestions for Authors

1. In the introduction, line 59 to 73 describes the anatomy very well. But describing this paragraph along with a graphical summary will be helpful for the readers.

2. In the results, the figure subsection is not cited. In line 199 and line 211, the figure subsection should be mentioned clearly to understand the results.

3. In the method section, line 146 antibody lot number is mentioned and not cataloged, but the correct catalog number is mentioned in line 137 which is mentioned as porcine gastric mucosa pepsin.

4. Authors have used purified porcine pepsin in western blot detection and are assuming that pepsin molecular weight should be around 36 kDA. But if you check the antibody used which has used the recombinant pepsin 4 are detected at 42kDa. The submitted blot has a higher band which could be the real pepsin band. 

In the literature, I have found a paper that has done pepsin detection in saliva. This paper has also detected the pepsin in the saliva and the band is higher than 35 kDa. I would suggest in-gel trypsin digestion of the band to be sure that the pepsin detection is correct. Also, they can use peptest quantitative analyzer to quantify pepsin in saliva.

Comments on the Quality of English Language

none

Reviewer 2 Report

Comments and Suggestions for Authors

33 IHC-positive.         Abbreviation should be explained.

37 reflux symptoms index; reflux findings score

55 Belafsky and colleagues developed the Reflux Symptom Index (RSI) questionnaire and 

56 Reflux Finding Score (RFS) scale.            Should be references for RSI and RFS

 Table 1. Brodsky tonsil grading scale > 2.    Should be explained in the text

The 101 study included 76 pediatric patients.            It should be more information about participants – age , sex, weight, heigt, BMI percentiles, comorbidities, medications, family history for allergy, imunodeficiency, GERD and other diseases. Maybe there is connection with age or sex, weight, etc.

259  LPR has become significantly prevalent in the pediatric population over the past decades (3). There is no data about LPR in this article.

Discussion should be better.

American Gastroenterology Association (AGA) expert review on Extraesophageal GERD does not use the term LPR at all (Clinical Gastroenterology and Hepatology 2022;20:984–994). It should be discussed.

For how long stays pepsin in the tonsils, saliva ? Maybe before sampling the patient doesn‘t have GER ?

Comments on the Quality of English Language

No 

Round 2

Reviewer 1 Report

Comments and Suggestions for Authors

I have read the authors remarks on my comments, and I found them satisfactory. I understand the constrains of clinical samples and I found it acceptable that they can't do further investigation.